# Transcriptional Signatures and Network-Based Approaches Identified Master Regulators Transcription Factors Involved in Experimental Periodontitis Pathogenesis

**DOI:** 10.3390/ijms241914835

**Published:** 2023-10-02

**Authors:** Emiliano Vicencio, Josefa Nuñez-Belmar, Juan P. Cardenas, Bastian I. Cortés, Alberto J. M. Martin, Vinicius Maracaja-Coutinho, Adolfo Rojas, Emilio A. Cafferata, Luis González-Osuna, Rolando Vernal, Cristian Cortez

**Affiliations:** 1Escuela de Tecnología Médica, Facultad de Ciencias, Pontificia Universidad Católica de Valparaíso, Valparaíso 2373223, Chile; emiliano.vicencio@pucv.cl; 2Centro de Genómica y Bioinformática, Facultad de Ciencias, Ingeniería y Tecnología, Universidad Mayor, Santiago 8580745, Chile; josefa.nunezb@mayor.cl (J.N.-B.); juan.cardenas@umayor.cl (J.P.C.); 3Escuela de Biotecnología, Facultad de Ciencias, Ingeniería y Tecnología, Universidad Mayor, Santiago 8580745, Chile; 4Departamento de Biología Celular y Molecular, Facultad de Ciencias Biológicas, Pontificia Universidad Católica de Chile, Santiago 8331150, Chile; cortes.bastian@gmail.com; 5Laboratorio de Redes Biológicas, Centro Científico y Tecnológico de Excelencia Ciencia & Vida, Fundación Ciencia & Vida, Santiago 7780272, Chile; proteinomano@gmail.com; 6Escuela de Ingeniería, Facultad de Ingeniería, Arquitectura y Diseño, Universidad San Sebastián, Santiago 8420524, Chile; 7Centro de Modelamiento Molecular, Biofísica y Bioinformática, Facultad de Ciencias Químicas y Farmacéuticas, Universidad de Chile, Santiago 8380492, Chile; vinicius.maracaja@uchile.cl (V.M.-C.); adolfo.rojas@ug.uchile.cl (A.R.); 8Advanced Center for Chronic Diseases—ACCDiS, Facultad de Ciencias Químicas y Farmacéuticas, Universidad de Chile, Santiago 8380492, Chile; 9Laboratorio de Biología Periodontal, Facultad de Odontología, Universidad de Chile, Santiago 8380492, Chile; emilio.cafferata@upch.pe (E.A.C.); luisgodont@gmail.com (L.G.-O.); rvernal@uchile.cl (R.V.)

**Keywords:** periodontitis, chronic inflammation, gene expression, transcriptome, gene regulatory networks, master regulators transcription factors

## Abstract

*Periodontitis* is a chronic inflammatory disease characterized by the progressive and irreversible destruction of the periodontium. Its aetiopathogenesis lies in the constant challenge of the dysbiotic biofilm, which triggers a deregulated immune response responsible for the disease phenotype. Although the molecular mechanisms underlying periodontitis have been extensively studied, the regulatory mechanisms at the transcriptional level remain unclear. To generate transcriptomic data, we performed RNA shotgun sequencing of the oral mucosa of periodontitis-affected mice. Since genes are not expressed in isolation during pathological processes, we disclose here the complete repertoire of differentially expressed genes (DEG) and co-expressed modules to build Gene Regulatory Networks (GRNs) and identify the Master Transcriptional Regulators of periodontitis. The transcriptional changes revealed 366 protein-coding genes and 42 non-coding genes differentially expressed and enriched in the immune response. Furthermore, we found 13 co-expression modules with different representation degrees and gene expression levels. Our GRN comprises genes from 12 gene clusters, 166 nodes, of which 33 encode Transcription Factors, and 201 connections. Finally, using these strategies, 26 master regulators of periodontitis were identified. In conclusion, combining the transcriptomic analyses with the regulatory network construction represents a powerful and efficient strategy for identifying potential periodontitis-therapeutic targets.

## 1. Introduction

Periodontitis is a long-term immune-inflammatory disease that damages the periodontium, causing soft tissue breakdown and tooth-supporting alveolar bone resorption [1]. This pathology is considered the most prevalent osteolytic disease in humans and one of the most common causes of tooth loss in adults, affecting more than 11% of the worldwide population [1,2]. The clinical signs of periodontitis include gingival inflammation, clinical attachment loss, and the development of a periodontal pocket [3]. Periodontitis has been associated with the pathogenesis of several other inflammatory diseases, systemic conditions, and extra-oral comorbidities, such as cardiovascular disease, diabetes mellitus, rheumatoid arthritis, obesity, osteoporosis, atherosclerosis, adverse pregnancy outcomes, cancer, inflammatory bowel disease (IBD), nonalcoholic fatty liver disease, and Alzheimer’s disease, among others [4,5].

Dysbiotic changes in the diversity and relative abundance of the bacterial communities that colonize and inhabit the subgingival sulcus are involved in the etiopathogenesis of periodontitis. The persistent presence of the dysbiotic subgingival biofilm promotes and leads to an exacerbated and dysregulated host immune response, which is responsible for the destructive clinical features of the disease [6,7]. The persistent inflammatory response orchestrated by the cytokines and chemokines released during this process leads to the recruitment of leukocytes and the development of chronic periodontitis, whose hallmark is the pathologic resorption of the tooth-supporting alveolar bone [8,9]. Bone remodeling is the predominant metabolic process regulating the bone structure and function, with the osteoclast being the major modulator of these processes. Several factors modulate and regulate remodeling processes, among which the RANK/RANKL/OPG intercellular protein system is critical for the activation and differentiation of osteoclasts [10]. RANKL (Receptor activator of nuclear factor-kappa B ligand) promotes osteoclastogenesis through the activation of RANK (Receptor activator of nuclear factor-kappa B). This mechanism is modulated by osteoblasts, which produce and secrete osteoprotegerin (OPG), a decoy receptor for RANKL with an osteoprotective role. OPG blocks RANKL signaling through its cognate receptor, RANK. The RANKL/OPG ratio imbalance is thought to deregulate bone remodeling, driving bone loss when RANKL concentrations exceed OPG relative to normal physiology [11]. Thus, osteoclastogenesis and the subsequent pathological resorption of tooth-supporting alveolar bone depend on the amount of RANKL in the periodontitis microenvironment [12,13].

Even though the molecular processes involved in the pathogenesis of periodontitis have been extensively studied, it is still unknown what regulatory mechanisms govern these processes at the transcriptional level. Different omics technologies have benefited personalized medicine by enabling simultaneous and in-depth whole genome molecular analysis in response to environmental cues at any given time [14,15]. In particular, in the numerous in vitro and in vivo studies that have contributed to a partial understanding of the molecular mechanisms underlying the pathogenesis of periodontitis, the most frequently utilized technique has been the transcriptome microarray [16,17,18,19,20,21,22,23,24,25]. In this study, we employed high-throughput RNA sequencing (RNA-Seq), the standard approach for understanding the changes in RNA levels underlying cellular responses induced by environmental perturbations [26]. Contrary to microarrays, RNA-Seq enables unbiased, non-targeted transcript detection, making it a technology with numerous practical advantages for the early detection of new transcripts, allelic variants, non-coding RNAs, and changes in lowly expressed genes, including those that encode proteins that control gene expression, such as transcription factors [26].

Due to the functional interdependencies between the various molecular components of a cell, diseases are never caused by variations in a single gene but rather by disruptions in the intricate intracellular and intercellular networks that connect the various tissues [27]. Integrating multiple types of omics high-throughput experiments has produced extensive molecular interaction maps, providing an integrated overview of diverse regulatory programs [28,29,30]. Nonetheless, there are no experimental data for most contexts of interest, i.e., diseases and regulatory interactions are often inferred from expression data solely [31,32]. Therefore, one of the currently employed approaches to identify specific regulatory interactions that will allow a better mechanistic understanding of complex human diseases to find disease-related genes relies on examining transcriptomic data [33,34,35]. Using network-based approaches to determine disease-associated biological interactions has helped simplify the complexity and heterogeneity of biological systems by using specific computational methods [33,36,37]. The networks representing transcriptional regulation in whole biological systems are gene regulatory networks (GRNs). GRNs often play a crucial role in identifying critical regulators of different types and the relationships between them, ranging from transcription factor (TF)-encoding gene and non-coding RNA (ncRNA) regulatory interactions in various cellular processes and signaling [38,39]. Transcriptomic expression data, molecular interaction networks used to build GRNs, and analysis methodologies that infer context-specific interaction networks have made it possible to compare physiological and pathological contexts to find potential early disease markers and master regulatory candidate genes in various pathological conditions, including periodontitis [29,40,41,42,43,44]. However, there are no reports of massive sequencing to analyze transcriptomic data and create GRNs explaining how transcription is regulated during experimental periodontitis.

Because of the diversity of molecular processes and the numerous signaling pathways involved in the pathobiology of periodontitis, this study aims to report the complete repertoire of differentially expressed transcripts during experimental periodontitis, construct GRNs, and identify master regulator (MR) TFs in this specific context of the disease. For this, we generated a model of ligature-induced experimental periodontitis that recreates the destructive inflammatory conditions of human periodontitis in mice. Following disease induction, we performed a massive sequencing (RNA-Seq) of the palatal mucosa of healthy and periodontitis-affected mice. This allowed us to conduct a network analysis based on detecting co-expression modules and regulatory interactions. After identifying groups of genes that follow similar expression patterns and their transcriptional regulators, we used our GRNs to disclose the MRs driving periodontitis. Our approach yielded results consistent with what is already known about this disease and helped discover new potential periodontitis-therapeutic targets.

## 2. Results

### 2.1. Experimental Model Confirmation: Alveolar Bone Loss

The dysbiosis of resident oral bacterial communities is a critical requirement in the pathobiology of periodontitis [5]. The induction of experimental periodontitis using the ligation method allows for recreating the destructive inflammatory conditions associated with the onset and progression of human disease in animal models [45]. Its main advantage is that the disease can begin at a predetermined time and cause pathological bone resorption rapidly. Similar to human periodontitis induced by subgingival biofilm dysbiosis, ligatures facilitate accumulation and local bacterial penetration, which, in turn, promotes inflammation and alveolar bone loss. For this reason, this model has been widely utilized for mechanistic studies and assessing the potential of new periodontitis therapeutic strategies [12,45] (Figure 1A). Because the resorption of the tooth-supporting alveolar bone is the hallmark of periodontitis, the maxillary molars of non-ligated mice and ligature-induced periodontitis mice were analyzed using micro-CT to assess the experimental model. A 3D reconstruction (Figure 1B), and the subsequent measurement of four trabecular bone structural parameters (Figure 1C–G), revealed extensive tooth-supporting alveolar bone resorption between the mesial surface of the first molar and the distal surface of the third molar in mice subjected to experimental periodontitis, 15 days after ligation (Figure 1B). The molecular parameters that drive bone dynamics and pathogenic bone resorption were also evaluated. It has been very well documented that the imbalance of the RANKL/OPG ratio deregulates bone remodeling, leading to bone loss when RANKL concentrations exceed OPG relative to normal physiology [13]. Confocal immunofluorescence images showed extensive RANKL expression and low OPG expression in biopsies from ligature-induced periodontitis mice (Figure 1H). In addition, we examined the RANKL and OPG expression in mice periodontal tissues using qPCR. A significant increase in both transcripts was evidenced in periodontitis-affected mice compared to non-ligated mice (Figure 1I,J). However, consistent with alveolar bone loss, periodontitis-affected mice showed a significantly higher RANKL/OPG ratio (Figure 1K). The results in Figure 1 indicate that experimental periodontitis was successfully achieved.

### 2.2. Global Expression Patterns and Differential Gene Expression (DGE) Analysis in Experimental Periodontitis

After ensuring that experimental periodontitis was successfully caused, we evaluated the transcriptional changes by performing complete transcriptome RNA sequencing (RNA-Seq) on the whole palatal mucosa of healthy and periodontitis-affected mice. The global transcriptional changes related to experimental periodontitis evidenced 366 protein-coding genes and 42 non-coding genes that were differentially expressed (DE) (Figure 2A). The criterion applied for this analysis: only genes with a significance level of p.adj < 0.05 and a fold change > 1.0 in periodontitis-affected mice compared to healthy mice were considered differentially expressed. According to these thresholds’ magnitude of change, we identified 313 upregulated and 95 downregulated genes in response to periodontitis induction (Figure 2B). Figure 2C depicts the biological variability among the samples as heatmaps with hierarchical clustering and expression levels of protein-coding genes differentially expressed. This analysis allowed us to identify clusters of genes with different expression patterns according to the heterogeneity of both experimental groups (Figure 2C). Then, to delve into the functional pathways that underlie the differential expression of these transcripts, we performed an enrichment analysis of the functional terms from Gene Ontology (GO) and the Elsevier Pathway Collection. This analysis revealed an overrepresentation of the functional terms of the proteins involved in periodontitis, specifically those related to the immune response. (Figure 2D). Moreover, consistent with the etiopathogenesis of periodontitis, our analyses revealed an enrichment of GO terms related to a variety of inflammatory disorders and those associated with extracellular matrix remodeling and neutrophil activation, such as glomerulonephritis, ichthyosis vulgaris, atopic dermatitis, myocarditis, and cathepsins in periodontitis (Figure 2D). In particular, the differential expression analysis showed that the genes *IL-1A*, *POSTN*, *IL-6*, *MMP13*, *CTSK*, *SERPINE1*, *C5AR1*, *NLRP3*, *TIMP1*, *CCR5*, *MMP8*, *MMP9*, *ELN*, *TFPI2*, *FN1*, *THBS2*, *MSR1*, *APLN*, *SELP*, *CCL8*, *CCL7*, *S100A9*, and *S100A8* are part of the most over-represented pathway in the transcriptome of periodontitis-affected mice (Figure 2E).

### 2.3. Gene Modules Associated with Periodontitis

Next, we examined modular gene co-expression networks, which effectively identify system-level functionality using transcriptomic data. This analysis allows for the functional classification of genes and the identification of genes with regulatory activity linked to each module [46,47,48,49]. Consequently, this analysis aims to identify genes that may interact or are similarly regulated during the pathology. We interrogated our data with CEMiTool [50] to identify groups of genes with similar expression patterns in periodontitis and control groups. Comparing both experimental groups, we identified thirteen co-expression modules (Figure 3A,B) with varying representation degrees and gene expression levels, modules M1 to M13. The M1 and M2 modules were the most enriched, with 1.252 and 400 genes upregulated in periodontitis, respectively (Figure 3A). Functionally, M1 and M8 revealed enrichment in the pathways associated with activating and regulating the immune response (Figure 3B and Figure 4). The other modules evidenced enrichments in the functional terms related to various functions, such as the structure, reorganization, and dynamics of the extracellular matrix (M2 and M10); muscle contraction, dynamics of actin, myosin, and intermediate filaments (M3, M6, and M12); and endothelial, epithelial, and epidermal development, proliferation, and morphogenesis (M4 and M11), among others (Figure 3B, Figure 4 and Figure A1). CEMITool additionally incorporates a combined co-expression analysis with protein–protein interaction data. As a result, the expression of the essential genes involved in immune regulation, such as Forehead box P3 (*FOXP3*), the Cluster of Differentiation 4, 40, and 28 (*CD4*, *CD40*, and *CD28*), IL-1b, macrophage inflammatory protein 2 (*CXCL2*), and C-X-C motif chemokine ligand 9 (*CXCL9*), were identified as hubs, nodes with the greatest number of connections, in modules M1 and M8 (network of Figure 4A,C). On the other hand, the hubs in module M2 (network in Figure 4B) were the genes Paired box 6 (*PAX6*), Actin alpha 1 (*ACTA1*), Caveolin 3 (*CAV3*), Myogenic Differentiation 1 (*MYOD1*), Progesterone Receptor (*PGR*), and Cyclin-dependent kinase inhibitor 1C (*CDKN1C*), among others.

### 2.4. Context-Specific Gene Regulatory Network on Experimental Periodontitis

Following this, we analyzed gene regulatory network GRNs focused on identifying the regulators causing increased or decreased gene transcription levels, thus altering the amount of proteins present, which may impact particular cellular functions. Even though regulatory proteins, such as transcription factors, are involved in many biological or disease processes, which TF is involved in the origin or affected by the disease etiology has not been fully identified yet [51]. We performed a differential network analysis to identify and characterize the complete repertoire of transcription factors involved in periodontitis and to reveal how these genes work together to produce this phenotype. We constructed context-specific GRNs using the normalized expression count matrix obtained from the expression analysis and high-quality confidence TF–target interactions following the procedure previously reported in Martinez-Hernandez et al. (2021) [42]. We generated two context-specific GRNs: healthy and periodontitis. We next used *LoTo*, which is a method that finds and identifies local topological variations between context-specific GRNs and generates a combined network that was further analyzed in Cytoscape (Figure 5A).

The largest network, which combines the two compared contexts, comprises 6.263 nodes, of which 761 are TFs, and 19.220 connections (Figure A2). A subnetwork was built from this network, selecting nodes that belonged to the modules identified with CEMiTool in the co-expression networks (Figure 3, Figure 4 and Figure A1). The subnetwork includes 12 clusters of genes from 12 clusters (module M13 was not present in the GRN) and their regulatory interactions; it is composed of 166 nodes, 33 TFs, and 201 connections after removing the 413 nodes that were not connected to any other nodes in any of the modules (Figure 5B). Module 1 is associated with the host immune response and is the largest group of nodes in the subnetwork. Notably, there are three transcription factors expressed only in periodontitis conditions (yellow nodes) in module 1, *SPIC*, *FOXG1*, and *OTX2*, and the TF *OLIG1* is only present in the control network. Remarkably, when selecting the first neighbors of these four TFs (Figure 5B and Figure A3A), there are two genes DE over-expressed in the disease model, *SERPINE1* and *DCLK1*, both part of module 2. Regarding the eight TFs coding genes in module 2, there are only three present in the disease condition, *SOX11*, *NR1H4*, and *EN1*, six are over-expressed in the disease (*SOX11*, *NR1H4*, *MYOG*, *ANKRD1*, *MMP9* and *TBX18)*, and only *MSX2* is over-expressed in the control samples. Notably, when selecting the first neighbors of these eight TFs in the general network, there are four TFs shared with the equivalent subnetwork of module 1, *YY1*, *MEF2C*, *SMAD3*, and *NFKB1*. In addition, there are two DE genes over-expressed in this subnetwork of the TFs in module 2: *MYL2* and *MYH7B*, both coding for different isoforms of myosin and, respectively, part of module 6 and module 12. Interestingly, there are 85 DE genes in module 2 that are also part of the GRN, 3 of them over-expressed in the control (*CPNE5*, *PLIN1* and *MYOC*) and the other 82 in periodontitis, which are enriched in inflammatory response GO terms (Figure 5B and Figure A3B).

Module M3 has 30 genes in our GRN, 8 of which are TFs, with *ARX* present only in the disease and *PAX6* only present in the control condition. *PAX6* regulates genes in the same module as well as some gene modules M2 and M1. *PAX6* and its regulatory interactions are only present in healthy conditions, suggesting the regulatory activity of this TF is lost in the disease state. Another member of Module 3, also regulated by *PAX6*, is *CRYBB1*, a gene that is over-expressed in periodontitis. There are another seven DE genes in Module 3, *PGR* (Progesterone receptor, a TF), *IL1A*, *SERPINB2*, and *IL12RB2* over-expressed in control and *SYN2*, *KCNG4*, and *B3GAT1* with expression levels significantly above in the disease-associated samples (Figure 5B and Figure A3C). M4 has only three nodes, none of them coding TFs in our GRN. Module 5 has 28 genes in our GRN, two of them code for TFs, and none is only present in one of the conditions; however, there are eight DE genes in this module. *BATF3* (a TF), *FLT3*, *S100A9*, *IGHV1-64*, *IGHV1-42*, and *IGHV1-34* are over-expressed in disease while *GPA33* and *VIL1* in the control. Similarly, to Module 5, no other TFs are present in only one condition in the remaining modules, nor are DE-TFs coding genes. Interestingly, the *SPI1*-TF stands out in the M8 module, a module associated with the inflammatory response, over-expressed in the disease samples. *SPI1* regulates several genes in the M1 module, which is involved in the immune response, and M2, which is related to metabolic and matrix activities (Figure 5B). It is important to highlight that this analysis allowed us to identify relationships between the modules based on the existence of regulatory interactions between TFs and their targets. Notably, merging the subnetworks, formed by the neighbors in first grade of TFs only present in one condition (only found in Modules M1, M2, and M3), generates a single component network with nodes from another three modules, which indicates the existence of a functional relationship between the modules. Moreover, selecting first-grade neighbors for all nodes in modules generates a large, single-component subnetwork of 1.565 genes (Figure A4) and 7.676 edges; only two pairs of TF genes are not linked to the main component (red arrowhead in A4), 441 of which code for TFs.

To conduct a more in-depth analysis of our data, we selected only the nodes in the health and periodontitis conditions from the list of all the genes that appear in the modules shown in Figure 5B. This approach showed that the PAX6 transcription factor is a central regulator of the health-related network, composed of 10 genes (Figure 5C), and that SIX3 is both regulated by and a regulator of *PAX6* (pink and black edges in Figure 5C). Furthermore, the regulatory interaction between *SIX3* and *PAX6* is present in both conditions in the subnetwork associated with health (black edge in Figure 5C). Importantly, our analysis revealed the loss of *PAX6* and its regulatory interactions during periodontitis (Figure 5D). In line with our findings, numerous neurodevelopmental, ocular, oral, and metabolic diseases have been linked to *PAX6* deficiency, mutations, and genetic variants [51,52,53,54,55,56,57].

We next only focused on the periodontitis-related nodes that were selected in the subnetwork shown in Figure 5C and are among those with the greatest variation in their local vicinity according to the F1 measure from *LoTo* (F1 < 0.99). This metric indicates if the neighborhood of a gene in a GRN varies when comparing two networks that represent two different network contexts, with lower values indicating greater changes in the GRNs. The subnetwork showed 16 genes isolated one from another, of which 6 correspond to TFs that are over-expressed in the disease model (Figure 5D). Based on the essentiality of bone dynamics during periodontitis, our analysis showed that the *PITX3*, a TF consistently more expressed in periodontitis, regulates *EN1*, a periodontitis-related TF that has critical functions during craniofacial osteogenesis and is also over-expressed in the disease model. Therefore, these findings indicate that their deficiency induces osteopenia in mice [58]. Furthermore, *EN1* regulates *FCER2A*, which is also overexpressed in periodontitis via the regulation of PAX5, up in the disease. *FCER2A* is also regulated by the *SPIC* periodontitis-associated TF. It is important to highlight that *FCER2* has been linked to several functions, such as cytokine signaling during inflammatory disorders [59] and *NOTCH2* activation [60]. This last TF, *NOTCH2*, positively regulates RANKL-induced osteoclast differentiation during bone remodeling [61]. 

Other targets of *SPIC* regulation in our subnetwork are CD5L and *MMP12*. The first is mainly upregulated and expressed by macrophages during inflammatory processes [62] whereas *MMP12* is involved in the breakdown of the extracellular matrix in physiological and pathological processes. The product of *MMP12* (elastase) degrades elastin and is highly expressed by macrophages and other stromal cells [63]. Figure 5D (gray circle) also reveals three independent relationships between different periodontitis-associated TFs. First, *NR1H4*, *SOX11*, and *OTX2* regulate differentially expressed genes such as *ABCB11*, *RP1*, and *DCLK1*, respectively. Furthermore, *FOXG1*-TF exerts upper regulatory relationships by modulating the periodontitis target genes *SERPINE1* and *DMRT3*.

### 2.5. Identification of Master Regulator Genes in Experimental Periodontitis

TFs designated as MR are essential in network system GRNs due to their hierarchical structure and extensive regulatory connections. These were first identified in *Drosophila* spp., where the regulatory activities of these genes were necessary and sufficient to direct specific developmental trajectories. Consequently, their absence substantially impacts the phenotype and topology of the network [64]. Then, we selected a set of “seed” genes (Figure 6A) from the CEMiTool-generated modules M1, M2, M3, and M8, along with genes from a subnetwork (Figure 6B) that regulated the expression of RANKL (*TNFSF11* gene) and OPG (*TNFSF11B* gene). The subnetwork was built by analyzing each module individually using the same protocol to identify MR TFs described in reference [65]. To accomplish this, the first and second neighboring nodes of each module’s selected seed genes were added in an undirected manner. Subsequently, we calculated the network metrics and centralities using Cytoscape. First, the subnetwork was filtered by erasing the nodes with both low indegree and outdegree ≤ 4. Next, we iteratively calculated indegree and outdegree after removing poorly connected nodes. This was repeated until we obtained a highly clustered subnetwork of TFs formed by nodes with indegree and outdegree greater than >3 (Figure 6A). From the master regulators of modules M1, M2, M3, and M8, as well as the genes that regulate the expression of RANKL and OPG, we identified 26 master regulator TFs of experimental periodontitis, constituting a subnetwork with 169 connections (Figure 6C and Figure 7).

Among these 26 MRs, Jun proto-oncogene (*JUN*), nuclear factor kappa B subunit 1 (*NFKB1*), and Spi-1 Proto-Oncogene (*SPI1*) have the greatest number of DEGs under their regulation, with 31, 25, and 18 target genes, respectively. In addition, MYC Proto-Oncogene (*MYC*), Transformation-Related Protein 53 (*TRP53*), E2F Transcription Factor 1 (*E2F1*), *NFKB1*, and *JUN* had the most connections (indegree + outdegree) within the subnetwork, with 503, 453, 425, 393, and 341 edges, respectively (Figure 7). Our analyses also revealed that the expression levels of 12 master regulators (*AR*, *RB1*, *TRP53*, *PPARG*, *EP300*, *MYC*, *LEF1*, *CTMMB1*, *CREBBP*, *CEBPA*, *GATA3*, and *RUNX2*) were consistently higher in the healthy condition (U statistic test between 0 and 5, see Section 4). In contrast, 8 MRs (*ESR1*, *CEBPB*, *FOS*, *BCL6*, *E2F1*, *SPI1*, *STAT3*, and *ETS1*) were consistently expressed at higher levels (U test statistic between 10 and 16) in the periodontitis condition. Consistent with the pathobiology of the disease, these MRs regulate gene expression in a variety of biological processes, including the cell cycle, cell growth, differentiation, senescence, the oxidative stress response, the immune response (inflammation, lymphocyte proliferation, and differentiation), and bone metabolism (osteoclastogenesis) [66]. Because each of the 26 MRs regulates multiple target genes, Figure 7 lists the top 50 target genes for each master regulator.

## 3. Discussion

Periodontitis, a chronic disease leading to the progressive deterioration of tooth-supporting tissues, results from a complex interplay of factors, including the presence of periodontopathogens and microbial dysbiosis, destructive inflammation, aging, immune status, race, and smoking, among others, contributing to its severity [67]. The clinical presentation and affected anatomical sites in periodontitis show considerable variability, leading to unique cases that demand personalized diagnosis and treatment approaches. This diversity in symptoms is paralleled by heterogeneous gene expression patterns linked to lesion severity and the accuracy of analysis methods [18,20,46,68].

To mitigate this, we analyzed cross-sectional gene expression data from the palatal mucosa of mice with experimental periodontitis. This study aimed to identify critical regulatory molecules in the pathogenesis of periodontitis as potential therapeutic targets. Although the murine oral microbiota lacks the periodontal pathogens responsible for the clinical characteristics of human periodontitis, which could be considered a limitation of our study, the combination of this model with next-generation sequencing (RNA-Seq) for transcriptomic analysis allowed us to identify the complete repertoire of genes with differential expression in the signaling pathways present in physiological (healthy) vs. pathological (periodontitis) conditions (Figure 2). We also discovered novel modular gene co-expression networks, notably identifying 13 co-expression modules, including 1.652 immune regulation genes in two major modules (M1 and M2), while extracellular matrix dynamics were upregulated in periodontitis. Based on our global transcriptomic data, we conducted an extensive analysis to construct gene regulatory networks (GRNs) comprising 761 crucial transcription factors that govern the global gene network in health and disease conditions (Figure A2). Our analysis included data from 6.263 genes (Figure A1), from which we could identify transcription factors that exert direct control over context-specific GRNs. 

Under normal conditions, we identified 11 genes (*PAX6*, *SIX3*, *NR2E1*, *SP8*, *GCK*, *MMP9*, *CRYAA*, *CRYBB1*, *IAPP*, *KRT12*, and *PCSK2*), with *PAX6* playing a pivotal role in establishing a regulatory axis influencing other genetic components (Figure 5C). In pathological conditions (Figure 5D), we detected 17 genes (*BDNF*, *PITX3*, *EN1*, *PAX5*, *FCER2A*, *SPIC*, *MMP12*, *CD5L*, *SOX11*, *RP1*, *OTX2*, *DCLK1*, *NR1H4*, *ABCB11*, *SERPINE1*, *FOXG1*, and *DMRT3*), including 6 exclusives to periodontitis (*EN1*, *SPIC*, *SOX11*, *OTX2*, *NR1H4*, and *FOXG1*). Notably, we uncovered 26 master transcriptional regulators linked to periodontitis (Figure 6C). These factors regulate immune response, extracellular matrix dynamics, cell cycle regulation, and bone biology, including 19 of the 23 genes shown in Figure 2E. This network-based approach enables the integration of extensive datasets, offering insights into global gene behavior rather than individual effects [27]. Importantly, our study provides new therapeutic alternatives beyond the current approaches.

TFs bind to DNA’s specific regulatory sequences within the nucleus of a cell, activating or repressing genetic transcription. Consequently, they are essential for coordinating gene expression and directing various biological responses, and their dysregulation is strongly associated with human diseases [69]. Despite the significance of TFs in disease, periodontal research has focused primarily on identifying differentially expressed genes and deducing their functions through associated signaling pathways [16,17,18,19,20,21,22,23,24,25,70,71,72,73,74]. Moreover, changes in TFs expression are not always accompanied by alterations in functional activity. It is plausible that the gene encoding this protein maintains the same expression level regardless of its activity level [75]. Consequently, the contrast between TF gene expression and functional activity remains undetectable via transcriptomic-based differential expression analysis, potentially resulting in a deficiency in understanding the regulatory functions of these molecules.

When we compared our analysis of regulatory networks with previous studies whose objective was to identify periodontitis-MRs, we found several differences and some similarities. For instance, in 2016, Sawle et al. [43] identified 41 MR-TFs by analyzing gingival tissue biopsies from healthy and periodontitis-affected individuals. Using transcriptomic data (microarray-ChIP-Seq techniques), the authors conducted an untargeted analysis focusing on TFs-regulon enrichment with the Gene Set Enrichment Analysis (GSEA) tool, emphasizing a rigorous statistical approach. Only 2 of the 41 MRs in their study were identified in our analyses. ETS Proto-Oncogene 1 (*ETS1*) (Figure 6 and Figure 7 [U-test 15]) is a highly conserved TF throughout evolution. Its influence covers various biological areas and is essential in regulating cellular responses to external stimuli, in the control of cellular senescence, and in vital immune system functions. Mainly, it plays a critical role in the differentiation and activation of T and B lymphocytes, in addition to the production of proinflammatory cytokines [76,77]. One of its essential functions lies in the modulation of angiogenesis. This influence is manifested through regulating genes that control the migration and invasion of endothelial cells in tumors with a high expression of *ETS1*. In this sense, its participation is intrinsic in regulating the immune response, and its involvement extends to the development of autoimmune diseases and the progression of certain types of cancer [78,79,80]. The other TF identified by Sawle et al. [43] is *IRF4* (Interferon Regulatory Factor 4), which is expressed in immune system cells such as lymphocytes, dendritic cells, and macrophages, and can regulate diverse functions, such as proliferation, apoptosis, and differentiation. Nevertheless, its primary activity involves B lymphocyte maturation and differentiation. Consequently, it carries out its central regulatory functions by activating the innate and adaptive immune responses and is considered by some reports to be an MR of human periodontitis [66,81]. Even though we identified this TF in our analyses as a component of module 1 in our GRN (Figure 5B), and confirmed that it is directly regulated by *PAX5*, a component of our periodontitis context-specific GRN (Figure 5D), it surprisingly did not meet our criteria for being considered a master regulator. This could be due to differences in sample type, sequencing method, and bioinformatics tools between studies.

The study by He et al. [44] was based on ChIP-seq datasets derived from the biopsies of patients with periodontitis. They construct a GRN using DEGs to investigate the underlying regulatory pathways. These authors identified 19 TFs, of which *EGR1*, *ETS1*, *IRF4*, *RUNX3*, and *XBP1* stood out as crosstalk genes with increased expression in disease. Within this group, both *EGR1* and *ETS1* played a key role in regulating the expression of crucial immune genes and forming the immunosuppressive microenvironment observed during the pathogenesis of periodontitis. As mentioned previously, in addition to *ETS1*, our analyses also identified *EGR1* (Early Growth Response 1) as MR in periodontitis (Figure 6C and Figure 7). *EGR1* is a multifunctional transcription factor that regulates cell growth, differentiation, and apoptosis. In fibroblasts, it drives the expression of a wide range of genes closely linked to wound healing, tissue remodeling, and fibrosis formation. Given its pleiotropic function, its deregulated expression is directly related to ischemic lesions, cancers of various types, inflammatory phenomena, atherosclerosis, cardiovascular pathogenesis in general, and fibrosis formation [82]. In summary, the diverse signals associated with tissue injury are potent inducers of *EGR1* expression and activity. This suggests and would explain its presence as a master regulator in a destructive chronic disease such as periodontitis. In the same context, our research revealed four MRs; three are within our main subnetwork (*ESR1*, *FOS*, and *RUNX2* in Figure 6C), while the fourth, *SP1*, belongs to the bone-metabolism-related subnetwork (Figure 6B). Another study examining the regulatory networks involved in periodontitis and the relationship between periodontitis and dyslipidemia also identified these same four MRs [83]. Interestingly, *ESR1* and *FOS* in our study also regulate the four DEGs associated with atherosclerosis (*SERPINE1*, *IL-6*, *MMP9*, and *IL1A* in Figure 2D,E). These findings suggest that these MRs may be involved in the systemic comorbidities frequently attributed to periodontitis. Recent studies utilizing single-cell sequencing (scRNA) technology in the periodontal tissues of periodontitis-affected or healthy individuals revealed the differential expression of the *BCL6*, *FOS*, and *JUN* MRs during osteoclastogenesis [84]. In addition, during periodontitis, increased expression levels of the *FOS* and *JUN* genes were observed in CD4^+^ T-cells, indicating their importance in cell activation. In addition, fibroblasts upregulating *RUNX2* showed the activation of pathways related to osteoblastic differentiation and bone formation, consistent with its known role in regulating metabolism and bone dynamics [84]. Interestingly, *RUNX2* was also identified as a regulator of cell differentiation of keratinocytes belonging to the gingival junctional epithelium. Taken together, these findings suggest that RUNX2 might have a role in maintaining the integrity of the dentogingival junction in healthy periodontium [85]. 

Maekawa et al.’s [70] study provided substantial data on the differential expression of distinct genes related to the destructive periodontal inflammation occurring in mice with ligature-induced periodontitis. Indeed, similarly to our study, increased expressed levels of the DEGs *CTSK*, *MMP*9, *TIMP1*, *MMP19*, and *CCL9* were detected. When examining the regulation of these genes, all of them are regulated by several of the TFs herein identified (Figure 7). Furthermore, the expression levels of *S100A8* and *S100A9* were also significantly increased, suggesting an association between these genes and the inflammatory destruction of tooth-supporting alveolar bone. Our analyses also showed that these genes exhibit differential expression (Figure 2E) and are subject to *E2F1* regulation (Figure 6 and Figure 7). Collectively, these findings highlight how the MRs identified in our study, including those mentioned above and those widely recognized and relevant to the pathogenesis of periodontitis, such as *NFkB1*, *RELA*, *STAT3*, and *GATA3*, among others, consistently demonstrate their role as regulators of a variety of essential processes associated with the periodontitis pathogenesis. This relationship pattern remains consistent across the various sample categories, transcriptomic methodologies, and bioinformatics applications employed in the diverse studies reviewed.

To date, periodontitis is an important public health problem, reaching a prevalence of 62% in adults. In addition, periodontitis contributes to the global burden of chronic noncommunicable diseases, having an important economic impact on the world’s healthcare systems [86,87]. Due to its impact on the general health and quality of life of diseased people, the design of effective therapeutic strategies is a constant concern in contemporary dentistry [88]. Today, treatment options for periodontitis span a broad spectrum, ranging from non-surgical approaches, such as scaling and root planning supplemented or not with systemic antibiotics, to periodontal surgeries and regenerative procedures [89]. If we consider the predisposing genetic component and the individual immune status, there is a need to develop more personalized and precise therapeutic alternatives, thereby optimizing therapeutic approaches. Considering this, the “omics” methodologies herein investigated permit the combination of large data sets to generate predictive models that facilitate the identification of genes with therapeutic potential, including periodontitis MR-TFs. Given that MR-TFs are central to the regulatory networks that coordinate the gene expression patterns, their therapeutic intervention has attracted interest in several human diseases [66]. An exemplary case is the transcription factor EB (TFEB), recognized as a master regulator of the autophagy-lysosomal biogenesis pathway, which stimulates cell clearance. TFEB has emerged as a therapeutic modulator in lysosomal storage diseases [90,91] and a promising target in the treatment of neurodegenerative disorders (ND), such as Alzheimer’s disease and Parkinson’s diseases [92,93]. Similarly, Forkhead Box O3 (*FOXO3a*) is regarded as a regulator in autoimmune disorders. Evidence shows *FOXO3a* modulates the immune response by targeting *NFkB*, *FOXP3*, and cytokine release [94]. Studies indicate that the loss of function of *FOXO3a* leads to severe arthritis and enhances the autoimmunity-related colitis severity in animals [94]. Furthermore, *FOXO3a* is downregulated in autoimmune-affected patients [94]. Indeed, *FOXO3a* levels negatively correlate with systemic lupus erythematosus activity and clinical phenotype [94]. Moreover, Kruppel-like factor 4 (*KLF4*) acts as an epigenetic modulator in the context of chronic kidney disease [95]. Interestingly, there are reports of various master regulators identified in our research, including *JUN*, *STAT3*, *NFkB*, *TRP53*, *FOS*, *E2F1*, and *MyC*, as modulators that affect cell proliferation, apoptosis, inflammatory activity, and tumor angiogenesis in various human cancer models [66,70,72]. Similarly, as functional components of the JAK-STAT pathway, the transcription factors *STAT3* and CCAAT/enhancer binding protein beta (*CEBPB*) modulate bone metabolism in osteolytic diseases [96,97]. Silent Information Regulator 1 (*SIRT1*) has also been identified as a master regulator in endocrine and metabolic diseases, such as hyperuricemia, diabetes, hypertension, hyperlipidemia, osteoporosis, and polycystic ovary syndrome [98].

In summary, the combination of periodontitis-standardized experimental models with molecular and computational strategies implemented in the present study, successfully identified the master regulators in the pathogenesis of periodontitis. These regulators represent potential therapeutic targets that, once validated through mechanistic studies, could be therapeutically addressed via pharmacological interventions or immunomodulation, resulting in more individualized treatments.

## 4. Materials and Methods

### 4.1. Experimental Animals of the Study

Experimental groups consisted of healthy wild-type C57BL/6 mice (8 animals in total), 6 to 8 weeks old weighing approximately 20 g. Mice were housed in separate cages and maintained under pathogen-free conditions in a controlled environment corresponding to a 12:12 h light/dark cycle at 24 ± 0.5 °C, an air renewal rate of 15-room vol/h, and 40 to 70% relative humidity. Throughout the study, mice were fed sterile standard chow and water ad libitum. The study was approved by the Institutional Animal Care and Use Committee (Protocol code: BIOPUCV-BA 686-2023) and conducted in accordance with the ARRIVE guidelines [99]. All the experiments were carried out following the recommendations of the American Veterinary Medical Association (AVMA) [100], and the guidelines approved by the Council of the American Psychological Society (1980) for animal experiments [https://www.apa.org/science/leadership/care/guidelines, accessed on 1 February 2022].

### 4.2. Experimental Periodontitis Induction and Analysis of Alveolar Bone Loss

Periodontitis was induced using 5–0 silk ligatures, which were tied bilaterally around the maxillary second molars without causing damage to the periodontal tissues [45]. As controls, non-ligated mice were used. Animals were randomly allocated into two groups with four mice in each group: control group, non-ligated, and periodontitis group. After 15 days of ligature, animals were euthanized with a single overdose of ketamine/xylazine anesthesia, and samples of maxillae and palatal periodontal tissues were collected for further analysis. The bone resorption analysis using micro computed tomography (micro-CT) was performed as reported by Cafferata et al. (2020) [101]. Hemi-maxillae were dissected to remove soft tissues, immersed in a 2.5% sodium hypochlorite solution for 12 h, and washed with 70% ethanol [EtOH], followed by 90% and 100% EtOH for 24 h. Samples were scanned using SkyScan 1272 micro-CT equipment (Bruker, Belgium) at 80 kV, 125 mA, with a rotation step of 0.3°, 360° around the vertical axis, and a voxel size of 9 µm. The three-dimensional (3D) digitized images were obtained using NRecom reconstruction software v.1.6.9 (Bruker, Belgium). The images were re-oriented in space using DataViewer software v.1.4.4 (Bruker, Belgium) to standardize the position. Finally, a region of interest (ROI) was created in the transverse plane using CTan software v.2.2.10 (Bruker, Belgium). For the ROI creation and subsequent analysis, the site mesial of the first molar and the distal side of the third molar was used as a reference. 

### 4.3. Indirect Immunofluorescence Confocal Assays

The periodontal tissues biopsies were fixed in 10% formalin pH 7.4 for 24 h and then demineralized in 5% EDTA (Sigma–Aldrich, St. Louis, MO, USA) for 60 days. The specimens were then processed for paraffin embedding, and serial 8 µm sections were prepared using standard histological protocols. After antigen retrieval, the specimens were immunostaining with an anti-RANKL mouse IgG monoclonal primary antibody (ab45039) and an anti-OPG rabbit IgG polyclonal primary antibody (ab183910) overnight at 4 °C, followed by an Alexa Fluor 488-conjugated anti-rabbit goat IgG (#A11008, Invitrogen, Waltham, MA, USA) polyclonal secondary antibody, and an Alexa Fluor 555-conjugated anti-mouse donkey IgG (#A48270, Invitrogen) polyclonal secondary antibody for 2 h at 4 °C containing 10 µg/mL DAPI (4′,6′-diamidino-2-phenylindole dihydrochloride, #D1306, Thermo Fisher Scientific, Waltham, MA, USA). Finally, samples were mounted using ProLong^®^ Gold antifade reagent mountant (#P36930, Life Technologies, Carlsbad, CA, USA) and analyzed in a Leica TCS SP8 confocal laser scanning microscope (Leica Microsystem, Wetzlar, Germany). The images were acquired by confocal microscopy, using a 10× objective and an oil immersion plan-apochromat 63x objective (numerical aperture 1.4) in a masked manner by a single examiner (C.C). The series of images obtained from confocal z-stacks were processed and analyzed using Leica LAS AF v.3.0 (Leica Microsystem 2012, Wetzlar, Germany) and Imaris software v.7.4.2 (Bitplane AG, Andor Technology), as described by Cortez et al. (2015) [102].

### 4.4. Total RNA Extraction

To optimize the concentration and quality of the molecules, total RNA from the periodontal tissues of control animals and those subjected to experimental periodontitis was extracted using a combination of the TRIzol™ protocols (#15596026, Invitrogen, Waltham, CA, USA) and PureLink™ RNA mini-Kit (#12183025, Invitrogen, Waltham, CA, USA). The samples were homogenized using the TissueLyser II equipment (QIAGEN). At the same time, the total RNA’s concentration, quality, and integrity were evaluated using the Infinite^®^ 200 PRO NanoQuant (Tecan, Männedorf) and Bioanalyzer (Agilent Technologies, Santa Clara, CA, USA), respectively. RNAs with a RIN (RNA integrity number) ≥ 7 were considered for analysis [See Appendix A]. 

### 4.5. cDNA Synthesis and Gene Expression Analysis Using Quantitative Polymerase Chain Reaction (qRT-PCR)

Before the expression analysis of the transcripts, cDNA was synthesized from the total RNA using the iScript™ cDNA Synthesis Kit (#1708891BUN, Bio-Rad, Hercules, CA, USA). Levels of target transcripts were analyzed on the StepOnePlus™ Real-Time PCR System (#4376600, Applied Biosystems, Foster City, CA, USA) using KAPA SYBR FAST qPCR Master Reagent (Kappa Biosystems, Sigma–Aldrich Corporation, St. Louis, MO, USA). The oligonucleotide sequences of the primers used in this study were the following: RANKL Forward (F) 5′-AGGCTCATGGTTGGATGTGG-3′; and Reverse (R) 5′-TCTGTAGGTACGCTTCCCGA-3′; OPG, (F) 5′-TTGCCTTGATGGAGAGCCTG-3′ and (R) 5′-TCCTCAGACTGTGGGTGACA-3′; rRNA 18S, (F) 5′-GTAACCCGTTGAACCCCATT-3′, and (R) 5′-CCATCCAATCGGTAGTAGCG-3′. Each sample was run in triplicate, and the relative amount of target transcripts was by the 2^−ΔΔCt^ method after normalization to the 18S mRNA levels.

### 4.6. RNA Library Preparation and Sequencing

Illumina sequencing was performed at Genoma Mayor, Universidad Mayor, Chile. First, total RNA extraction was submitted to digestion with DNase I to avoid contamination with genomic DNA. The RNA concentration was then determined using the Quant-iT^TM^ RiboGreen^®^ RNA Assay Kit (Life Technologies), and its integrity was assessed using an RNA 6000 pico chip on the Bioanalyzer 2100 (Agilent Technologies, Santa Clara, CA, USA). Next, RNA libraries were prepared with the Illumina TruSeq Stranded mRNA LT Sample Preparation Kit (Low-Throughput Protocol) according to the manufacturer’s protocol [See Appendix A]. Next, the RNA was fragmented with divalent cations at elevated temperatures. First-strand cDNA synthesis produced single-stranded DNA copies from the fragmented RNA by reverse transcription. After second-strand cDNA synthesis, the double-stranded DNA was end-repaired, and the 3′ ends were adenylated. Finally, the cDNA fragments were combined with universal adapters, and the PCR was implemented and used to produce the final sequencing library. After validating the library using the DNA 1000 chip on the Agilent Technologies 2100 Bioanalyzer, the samples were quantified using qPCR, pooled together in equal concentrations, and run on an Illumina HiSeq for 100 cycles of paired-end sequencing.

### 4.7. RNA Sequencing [RNA-Seq] Data Analysis

A quality control check was performed with FastQC and detected adapter sequences were removed with Fastp in automatic mode for adapter identification. Next, clean reads were processed with Hisat2 [45] for mouse genome mapping (GRCm39), followed by the read summarization process with featureCounts [103], using GENCODE vM27 for gene annotation [104]. Finally, differential expression analysis was performed using the R DESeq2 package [105], establishing log_2_fold-change ± 1 as differential expression thresholds and adjusted *p*-value (p.adj) of ≤0.05. From this analysis, the normalized read count matrix generated by DESeq2 was extracted for further analysis [See Appendix A]

### 4.8. Co-Expression Modules Analysis

The R package “CEMiTool” [50] was used in conjunction with the DESeq2 normalized RNA-seq expression matrix to generate co-expression modules and evaluate related pathways. We also created a protein interaction network obtained from STRING version 11.5 [106], selecting interactions with a combined score ≥ 0.7. This network was then used to define co-occurrence modules using CEMiTool, as with the co-expression data. For over-representation of functional terms in each module, a GMT of GO terms from Msigdb (PMID: 16199517) was added to the CEMiTool workflow.

### 4.9. Construction of Gene Regulatory Network

A reference gene regulatory network containing only high-confidence transcription factors genes was built by merging: DoRoThea [107], TRRUST [108], and RegNetwork [109]. This reference network was filtered to build context-specific networks using the normalized counts of each RNA-seq sample [110]. The filtering process is as follows: for each context (RNA-seq sample), regulations were maintained if they arise from a transcription factor in which genes are expressed more than 0 in at least one replicate, and average normalized read counts are more than 10 and removed otherwise. In this way, we created gene regulatory networks for two contexts, health, and periodontitis, which were compared with *LoTo* [111] to identify network elements whose local topology varies between the two conditions. *LoTo* determines the presence or absence of network motifs in each context-specific network and outputs a file compatible with Cytoscape [112] for visualization [See Appendix A]. *LoTo* uses a color code to mark the nodes and edges present in both or only one of the networks compared. To select genes in this network generated by *LoTo*, we used those present only in one of the two contexts and in one of the modules assigned using CEMiTool [50].

### 4.10. Identification of the Periodontitis-Master Regulators

The identification of MR TFs is based on the definition of MRs described in Davis et al. (2017) [64]. Briefly, MR TFs are clusters of highly interconnected TFs that tend to interact physically between them and are not separated by many steps (network edges) from effector genes that are responsible for the phenotype of interest, or its change. We started using them as seed genes in modules highly related to the pathogenesis of periodontitis (M1, M3, and M8) and the genes coding for RANKL and OPG. We selected the regulator of the seed genes and the regulators of their regulators to define MR TFs. Following this, the subnetwork of TFs was filtered by erasing the nodes with both indegree and outdegree ≤ 4. Next, we iteratively calculated indegree and outdegree after eliminating poorly connected nodes until we obtained a highly clustered subnetwork of TF formed by nodes with both indegree and outdegree above 3. We further removed edges from the resulting subnetwork based on the absence of interactions reported in STRING database [105] between pairs of TFs. We finally removed TFs that did not meet the in- and outdegree criteria, leaving a subnetwork of 26 TFs connected by 169 regulatory interactions that are considered the MR that control the genes explaining the phenotypic differences between periodontitis and healthy mice.

## 5. Conclusions

The MR-TF genes identified in this study play a crucial role in regulating all pathogenic aspects related to the onset and progression of experimental periodontitis. Consequently, validating these genes in human clinical samples and subsequently modulating them in the periodontal tissue of individuals affected by periodontitis could offer novel and complementary therapeutic approaches, and their products could be used as potential biomarkers for the early detection of this prevalent disease. These perspectives can improve the quality of dental care by delivering more precise diagnoses, effective treatments, and personalized prevention strategies.

## Figures and Tables

**Figure 1 ijms-24-14835-f001:**
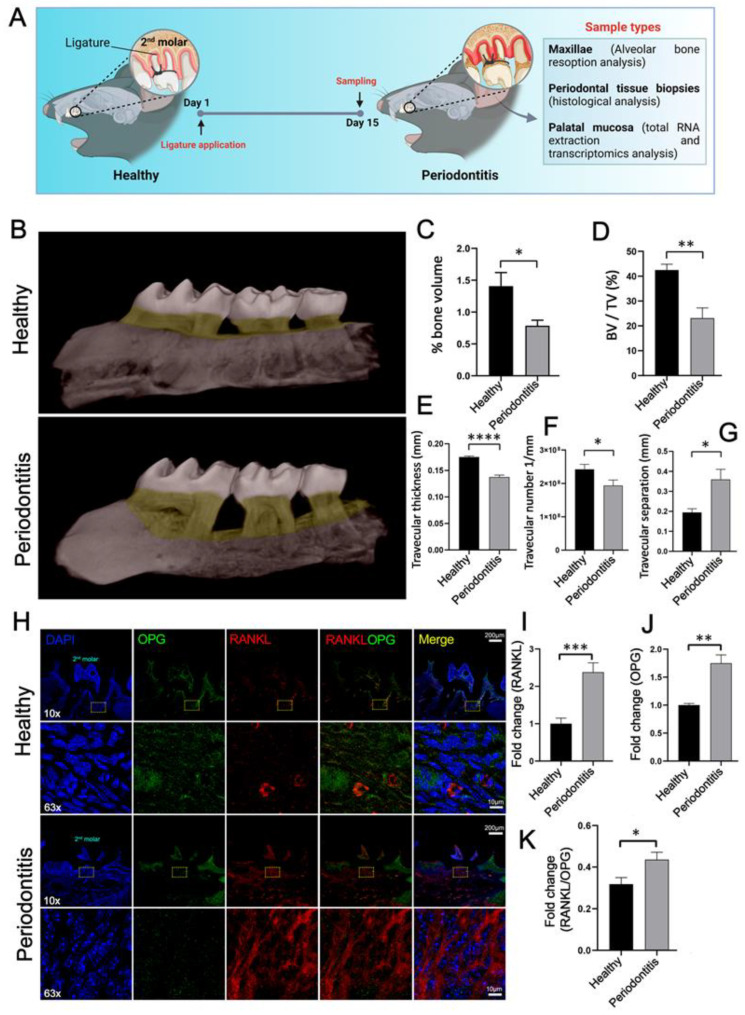
Experimental periodontitis and evaluation of alveolar bone resorption. (**A**) Schematic representation of ligature-induced periodontitis. At 15 days post-induction, the periodontal tissues indicated in the figure were extracted to carry out the analysis of this study. (**B**) Representative micro-computed tomography (micro-CT) 3D images of maxillae from mice with ligature-induced periodontitis and non-ligated mice used as healthy controls. Highlighted in yellow is the ROI described in the methods section, where the analyses of the structural parameters of trabecular bone were performed (**C**–**G**). (**C**) Alveolar bone volume quantified by micro-CT in maxilla specimens. Data are represented as the percentage of bone volume shown as mean ± SD (*n* = 4). (**D**) Bone volume fraction (BV/TV), which is expressed as a percentage and corresponds to the ratio between segmental bone volume and total volume. (**E**) Trabecular thickness, which is the mean thickness of the trabecula in millimeters. (**F**) Quantification of trabeculae number. (**G**) The trabecular separation, which is the average distance between trabeculae expressed in millimeters. These graphs show the average ± SD of four independent experiments. (**H**) Biopsies from mice with experimental periodontitis and from healthy controls were processed, immunostained, and analyzed using confocal microscopy. DNA-rich structures stained with DAPI; for the visualization of cell nuclei, osteoprotegerin (OPG) labeled with an anti-OPG antibody (in green); receptor activator of nuclear factor kappa-B ligand (RANKL) labeled with an anti-RANKL antibody (in red); and merge, which is the digital overlap of blue, green, and red channels shown in the fifth column. The yellow squares of the low-magnification (10×) images indicate the region of the periodontium where the images with high magnification (63×) were acquired, scale-bar 200 and 10 µm, respectively. Note the RANKL-increased expression in the biopsies from periodontitis-affected mice. Relative quantification of transcripts encoding for RANKL (**I**), OPG (**J**) proteins using quantitative real-time PCR normalized to the 18S rRNA transcript levels. (**K**) Ratio of the relative expression levels of the transcripts RANKL and OPG (RANK/OPG) in periodontal tissues (*n* = 4). * *p* < 0.05; ** *p* < 0.01; *** *p* < 0.001; **** *p* < 0.0001.

**Figure 2 ijms-24-14835-f002:**
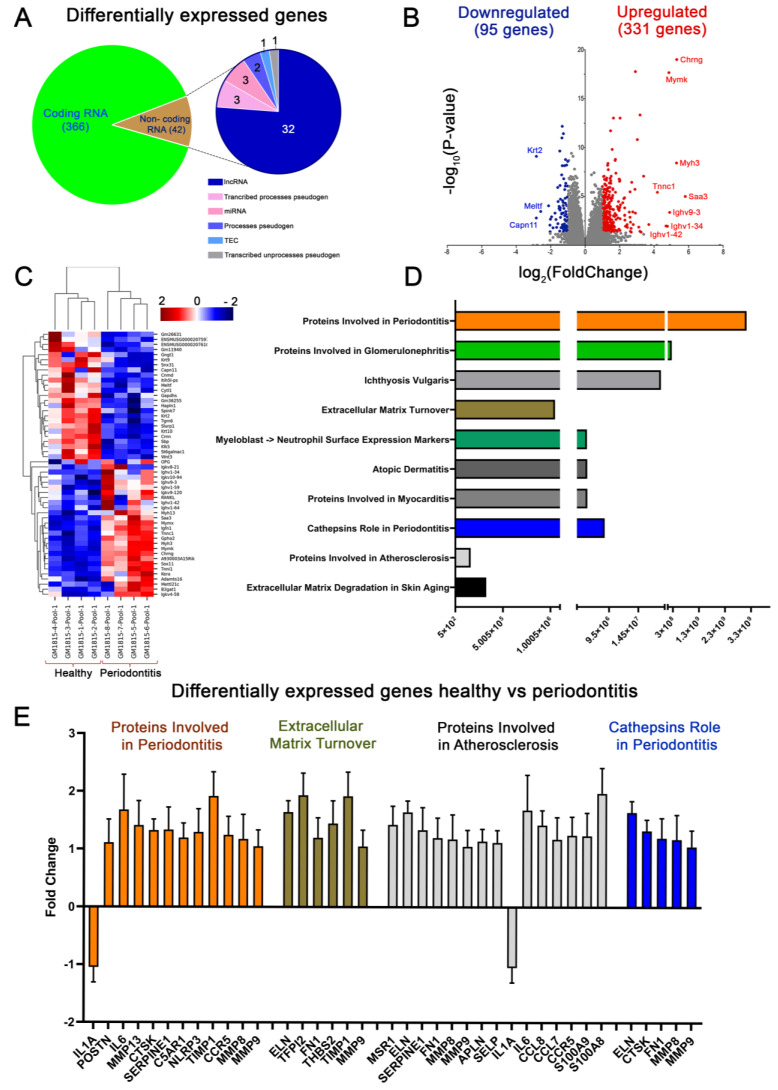
Transcriptomic analysis of mice with experimental periodontitis identifies changes in mRNA expression and pathology-associated pathways. (**A**) Differentially expressed (DE) genes from the transcriptomic analysis of the palatal mucosa of mice with experimental periodontitis. (**B**) Volcanic plot analysis derived from FPKM values showed DE genes (*p* < 0.05, Student’s *t*-test) in the palatal mucosa of mice with experimental periodontitis. (**C**) Heatmap representation of the RNA-Seq analysis of significant (*p* < 0.01, Student’s *t*-test) coding genes from the palatal mucosa of mice with experimental periodontitis. (**D**) The top 10 pathways significantly enriched for DE genes (*p* < 0.05, Student’s *t*-test) from the Elsevier Pathway Collection Database. (**E**) Expression profile of the enriched pathways’ DE genes from the Elsevier Pathway Collection Database. Differentially expressed (DE) genes from the transcriptomic analysis of the palatal mucosa of mice with experimental periodontitis.

**Figure 3 ijms-24-14835-f003:**
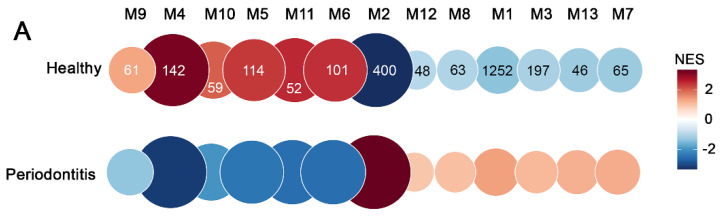
Analysis of co-expression modules in experimental periodontitis. (**A**) Gene set enrichment analyses that display the module activity. The normalized enrichment score (NES) is represented by the size and color of the circle, and the number in the circle shows how many genes belong to that module. (**B**) Representation of the 5 main biological processes in each of the 13 modules, using gene set enrichment analysis from MSigDB (PMID: 16199517) in CEMiTool.

**Figure 4 ijms-24-14835-f004:**
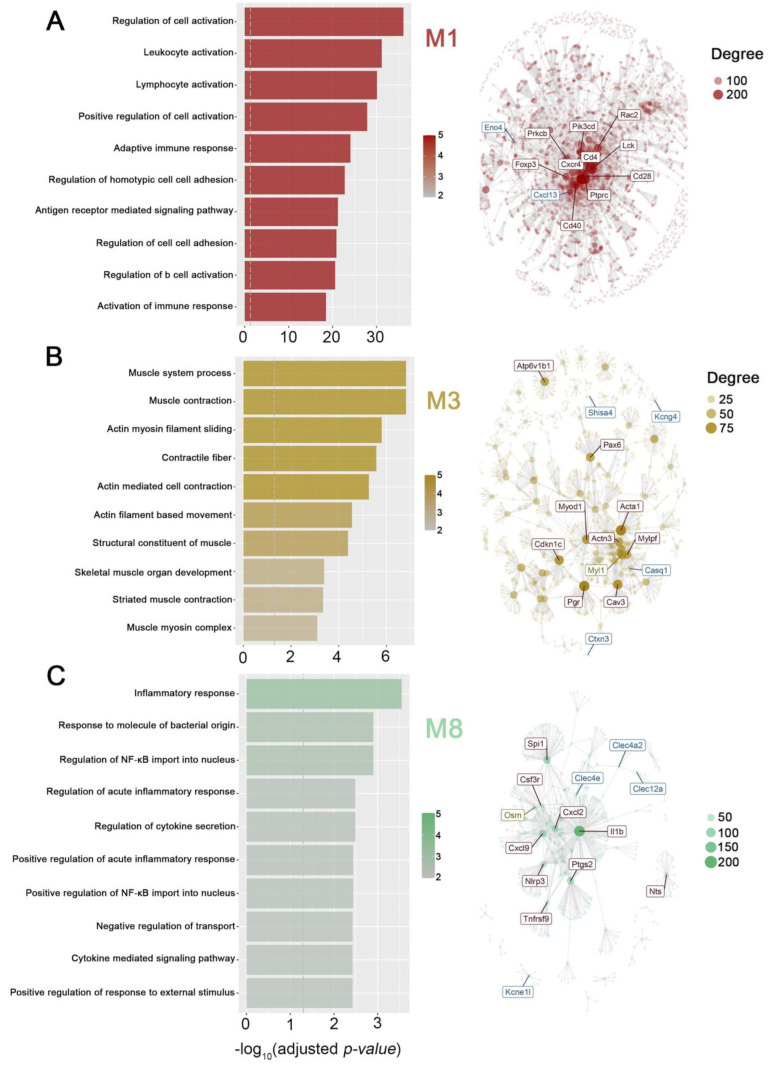
The most interconnected co-expression modules’ signaling pathways and gene networks. The top ten functional terms with the highest overrepresentation are displayed in a bar graph on the left. The core genes for each co-expression module are highlighted in the interaction plot on the right. Gene set enrichment analysis from Msigdb [PMID: 16199517] was used to conduct an overrepresentation analysis (−log_10_ adjusted *p*-value) on the modules M1 (**A**), M3 (**B**), and M8 (**C**) in the CEMiTool and String databases. The pathways were ordered by significance as indicated in the *x*-axis. The vertical dashed gray line indicates an adjusted *p*-value ≤ 0.05. The interaction plots of the most interconnected genes (hubs) are also highlighted for the M1, M3, and M8 modules, respectively. The size of the node is proportional to its degree.

**Figure 5 ijms-24-14835-f005:**
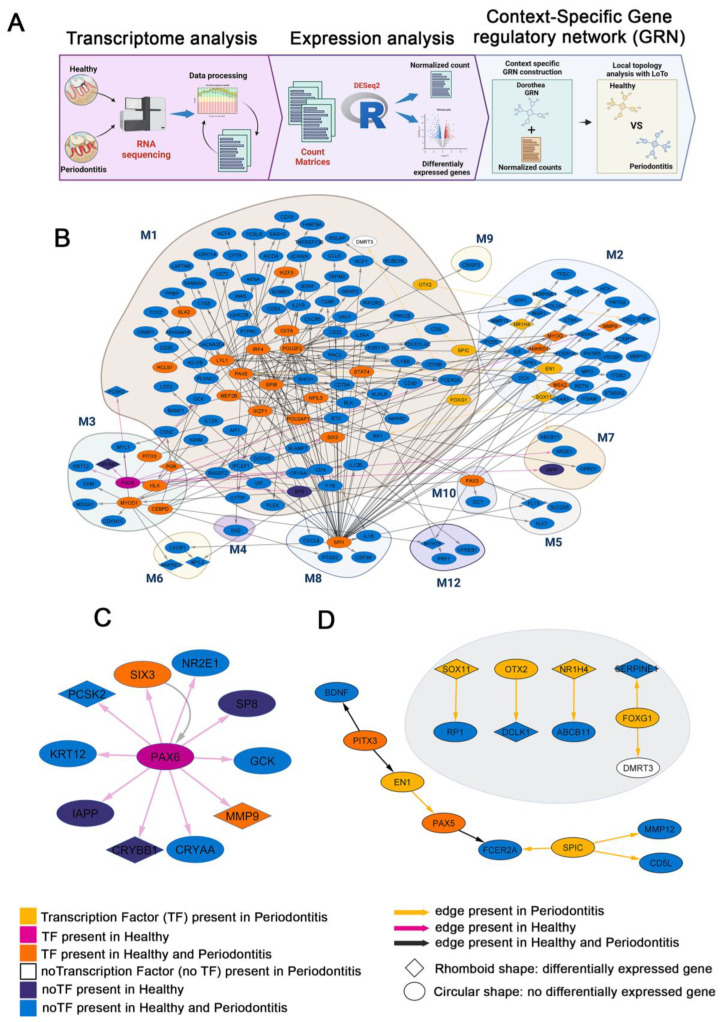
Construction of the gene regulatory network, according to co-expression modules of CEMiTool. (**A**) Pipeline to generate the GRNs for both the healthy and periodontitis contexts. First, RNA-seq data were obtained from experimental periodontitis induced previously. Following the processing of the raw reads, the read quality was evaluated using FastQC, adapters were removed using Fastp, and the data were mapped to the mouse genome using HIS2. Then, for the expression analysis, we applied the R package DESeq2. A log_2_ fold change ± 1 and a *p*-value ≤ 0.05 were used to determine whether a gene was differentially expressed. For GRN construction, first we normalized gene expression counts with DESeq2. We next used the expression of TF coding genes to filter a reference GRN by removing edges arising from non-expressed TFs, obtaining a healthy and a periodontitis network. Then, we used *LoTo* to identify local topological differences between context-specific GRNs and generate a combined network that marks topological network differences between the compared GRNs compatible with Cytoscape for visualization. (**B**) The subnet represents the co-expression modules obtained with the CEMiTool present in the GRN of Figure A2A. The subnetwork is divided into 12 gene modules and their regulatory interactions and consists of 163 nodes, 32 TFs, and 199 connections. Interaction subnetworks of TF with their first neighbors present under context-specific conditions. (**C**) Subnetwork in a healthy condition. (**D**) Subnetwork in an experimental periodontitis condition.

**Figure 6 ijms-24-14835-f006:**
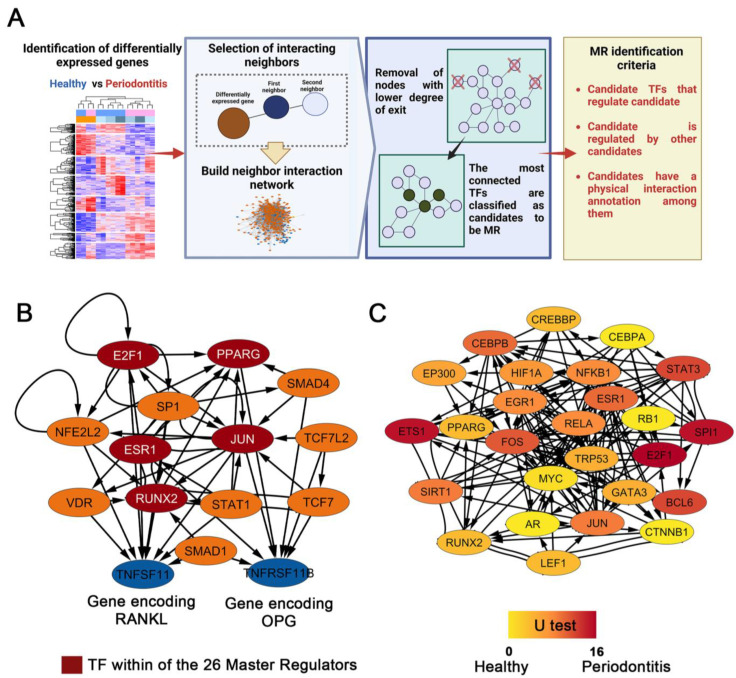
Construction of subnetworks to identify master regulator genes. (**A**) GRN pipeline for identifying master regulators. From our normalized gene count data, we selected the genes identified using CEMiTool belonging to modules M1, M3, and M8. The first and second neighboring nodes were added to these “seed” genes in an undirected manner, and then a subnetwork was built with all those nodes. Finally, all non-TF coding nodes and those without connections were removed, as specified in the method section. This allowed us to obtain a network with highly connected TFs. (**B**) Regulatory network built using seed genes encoding RANKL and OPG in mice. In dark red, the transcription factors that were identified as master regulators in 6C are shown. (**C**) Master regulator gene subnetwork. The genes in the regulatory subnetwork shown in 6B and modules M1, M2, M3, and M8 of CEMiTool were seed genes used to construct the subnetwork, which consists of 26 master regulators genes. The values of the U statistic exhibited as a color gradient (see Section 4 for details) indicate in which condition the master regulators are consistently more expressed (healthy or periodontitis) with intermediate values indicating no consistent increased expression in one of the conditions.

**Figure 7 ijms-24-14835-f007:**
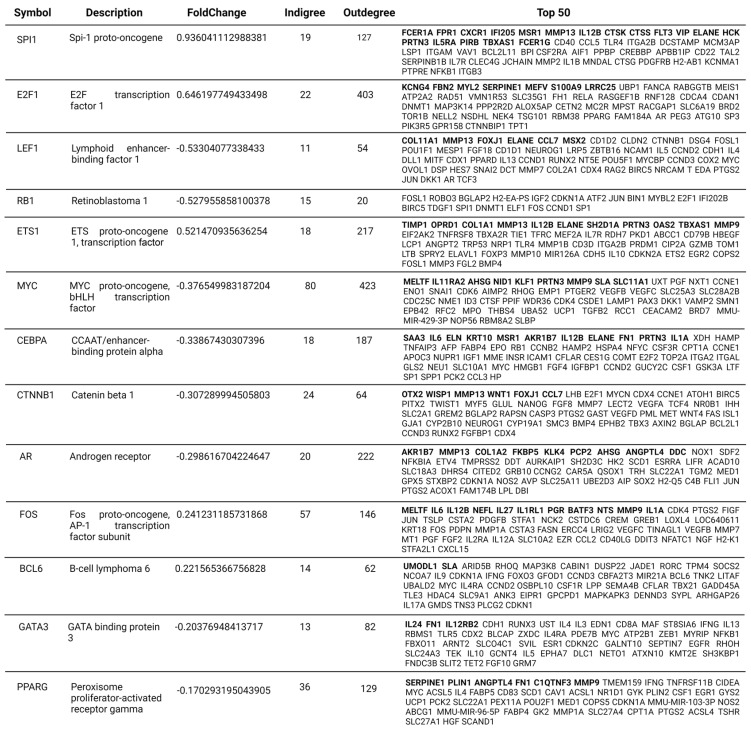
The top 50 genes regulated by the 26 MRs identified (Figure 6C). Highlighted in black are the DEGs for each.

## Data Availability

All sequence data are accessible with accession number BioProject ID: PRJNA1010977 https://www.ncbi.nlm.nih.gov/bioproject/1010977 (accessed on 31 August 2023).

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
