# Peer review of "Transcriptional Signatures and Network-Based Approaches Identified Master Regulators Transcription Factors Involved in Experimental Periodontitis Pathogenesis"

_ijms, 2023, doi:10.3390/ijms241914835_

Round 1
Reviewer 1 Report
This is a very thorough study that has generated a huge amount of data. In some cases the clusters of TFs and MRs are not surprising and build and delineate much of the data that has been generated by less sophisticated means.
Fig3 is a particularly good summary of the pathways so far elucidated.
The authors state 26 MRs were identified but it was not clear where this data was until page 15. Could the authors please change the title of Table 1 to read "The top 50 genes regulated by the 26 MRs identified " for clarity, and also emphasize Fig6c.
Similarly in a paper with so much information it is worth repeating which genes are included - see page 16

Author Response
Dear Reviewer:
First, we would like to express our sincere gratitude for your insightful and invaluable suggestions and comments. Your constructive review will undoubtedly contribute significantly to improving our manuscript. Next, we proceed to respond to your comments in the pdf attached:

Reviewer 2 Report
The paper is interesting and well written and organized.
however, some criticism arose from its content in some points, as follows:
- Conclusions should indicate the preliminarity of the results, which need confirmation on human models.
- If the animal model is sufficiently similar to the human one, the authors should state it in the first sentence of the results, explaining the induction of experimental periodontitis. Otherwise, they should report the study's limitations in the “discussion” or “conclusion” sections.
- Furthermore, experimental periodontitis, being “maintained under pathogen-free conditions,” lacks the microbial actors involved in the clinical environment. Could this one be a limitation of the study? Could the periodontal pathogens influence the expression of different genes/proteins involved in periodontitis? Please, discuss.
- the Discussion starts with the following sentence, which did not mention the role of peridontopathogen species :
“Periodontitis, a chronic disease leading to the progressive deterioration of tooth-supporting
tissues, results from a complex interplay of factors, including aging, immune status,
race, and smoking, among others, contributing to its severity””. Authors should consider them.
5. Page 10, first sentence: please, add the subject of the sentence “the…… is involved in the origin or affected by the disease etiology has not been
fully identified yet”
Author Response

(The authors gave the same response as above.)

Reviewer 3 Report
This manuscript reported the complete repertoire of differentially expressed transcripts during experimental periodontitis. It also constructs gene regulatory networks and identifies master regulator transcription factors in this specific disease context. The transcriptional changes revealed 366 protein-coding genes and 42 non-coding genes that were differentially expressed and enriched in the immune response. In addition, 13 co-expression modules were found with varying degrees of representation and gene expression levels. The gene regulatory network includes genes from 12 gene clusters, 166 nodes of which 33 encode transcription factors and 201 connections. Finally, 26 master regulators of periodontitis were identified using these strategies. The manuscript contains four keywords, six figures with different sections (Fig. 1 A to K; Fig. 2 A to E; Fig. 3 A and B; Fig. 4 A to C; Fig. 5 A to D; Fig. 6 A to C), four supplementary figures with sections (A1; A2 A and B; A3 A to C; A4), one table, and one hundred and seventeen references. Overall, it is a correct, very complete, and well-conducted paper.
General comments
This study highlights combining the transcriptomic analyses with the regulatory network construction represents a powerful and efficient strategy for identifying potential periodontitis-therapeutic targets. The study is methodologically well-established. The data management is appropriate according to the approach of the study. The results are well presented, being easy to read and interpret them. In the discussion section, the results of this study are adequately contrasted with those obtained by other researchers. A justifying explanation of the results is also provided. The manuscript also includes an appropriate concluding paragraph at the end of the discussion section.
Some further remarks are made on different sections of the manuscript.
Keywords
The manuscript presents four keywords. For keywords, where possible, please use Medical Subject Headings terms (MeSH Terms). Strictly, only “periodontitis” and “gene regulatory networks” are MeSH terms. An alternative MeSH term proposed could be “transcriptome” better than “transcriptomic”. Nevertheless, these suggestions about keywords are optional, not mandatory.
Other manuscript sections
Figure 3A is not cited in the text of the manuscript. Please cite it.
Please note that figure 6C (page 11 of 30) is cited before figures 6B (page 13 of 30), and 6A (page 13 of 30). Please order them in alphabetical sequence 6A, 6B, and 6C.
In text-reference number 103, please replace the parentheses "(103)" with square brackets "[103]".
References
Total number of the manuscript references: 117.
This is a complete and updated section. Nevertheless, the reference format does not match the journal’s reference format (ACS style guide). According to the journal’s guidelines, references should be described as follows, depending on the type of work:
Journal Articles:
1. Author 1, A.B.; Author 2, C.D. Title of the article. Abbreviated Journal Name Year, Volume, page range.
Books and Book Chapters:
2. Author 1, A.; Author 2, B. Book Title, 3rd ed.; Publisher: Publisher Location, Country, Year; pp. 154–196.
3. Author 1, A.; Author 2, B. Title of the chapter. In Book Title, 2nd ed.; Editor 1, A., Editor 2, B., Eds.; Publisher: Publisher Location, Country,Year; Volume 3, pp. 154–196.
Unpublished materials intended for publication:
4. Author 1, A.B.; Author 2, C. Title of Unpublished Work (optional). Correspondence Affiliation, City, State, Country. year, status (manuscript in preparation; to be submitted).
5. Author 1, A.B.; Author 2, C. Title of Unpublished Work. Abbreviated Journal Name year, phrase indicating stage of publication (submitted;accepted; in press).
Unpublished materials not intended for publication:
6. Author 1, A.B. (Affiliation, City, State, Country); Author 2, C. (Affiliation, City, State, Country). Phase describing the material, year. (phase:Personal communication; Private communication; Unpublished work; etc.)
Conference Proceedings:
7. Author 1, A.B.; Author 2, C.D.; Author 3, E.F. Title of Presentation. In Title of the Collected Work (if available), Proceedings of the Name of the Conference, Location of Conference, Country, Date of Conference; Editor 1, Editor 2, Eds. (if available); Publisher: City, Country, Year (if available); Abstract Number (optional), Pagination (optional).
Thesis:
8. Author 1, A.B. Title of Thesis. Level of Thesis, Degree-Granting University, Location of University, Date of Completion.
Websites:
9. Title of Site. Available online: URL (accessed on Day Month Year). Unlike published works, websites may change over time or disappear, so we encourage you create an archive of the cited website using a service such as WebCite . Archived websites should be cited using the link provided as follows:
10. Title of Site. URL (archived on Day Month Year)
For further information about the reference format proposed by the journal, please, consult the following link: https://www.mdpi.com/journal/ijms/instructions
Figures
Total number of the manuscript figures: 10 (6 figures and 4 supplementary figures). The figures have appropriate figure legends.
Tables
Total number of the manuscript tables: 1.
The table has an appropriate title and information. However, Table 1 features a font size that is too small, resulting in difficulty when reading. Please, consider increasing the font size for improved readability.
Author Response

(The authors gave the same response as above.)

Reviewer 4 Report
This manuscript entitled “Transcriptional Signatures and Network-Based Approaches identified Master Regulators Transcription Factors involved in Periodontitis Pathogenesis” has been reviewed sincerely. Authors showed 26 Master regulators-Transcription factors genes have important roles in pathogenesis of periodontitis using in vivo experimental periodontitis. I felt that the work was of interest, and well-performed. Although I have no serious comments for the submitted paper, there were several concerns as below.
Authors identified important 26 Master regulators-Transcription factors genes using software. Since these were imaginary candidate genes, the validity must be evaluated in this experimental models. Expression of several genes among 26 MR-TF genes should be valid.
Minor comments
1. The title should be revised. “Experimental periodontitis” may be important in this study.
2. Although the authors focused on TR genes, how about pro/anti-inflammatory cytokines such as IL-1, IL-10 in this model? Please discuss about this point.
3. Table 1 should be revised to be understood easily in readers of this paper.
Author Response

(The authors gave the same response as above.)

Round 2
Reviewer 4 Report
None